# ANCHORING ENTITIES: RETRIEVAL-AUGMENTED HALLUCINATION DETECTION

## ABSTRACT

Hallucination detection is crucial for large language models (LLMs), as hallucinated content creates significant barriers in applications requiring factual accuracy. Current detection methods mainly depend on internal signals like uncertainty and self-consistency checks, using the model's pre-trained knowledge to identify unreliable outputs. However, pre-trained knowledge may become outdated and has coverage limitations, especially for specialized or recent information. To address these limitations, retrieval-augmented generation (RAG) has emerged as a promising solution that grounds model outputs in external evidence. In this paper, we target a critical and practical learning problem *RAG-based hallucination detection* (RHD), where RAG is employed to enhance hallucination detection by addressing information updating challenges. To address RHD, we propose a novel method *Evidence-Aligned Entity Verification* (EAEV), which detects entity-level hallucinations by leveraging RAG to align generated entities with retrieved evidence contexts. Specifically, EAEV evaluates entity-evidence alignment through three complementary dimensions and introduces counterfactual stability analysis to ensure robust alignments under evidence perturbations. Experiments across multiple RAG benchmarks demonstrate that EAEV achieves consistent improvements over existing methods with strong generalization capabilities.

## 1 INTRODUCTION

The deployment of large language models (LLMs) in practical applications faces a critical challenge: models frequently generate factually incorrect or inconsistent content, known as hallucinations (Ji et al., 2023). This problem poses significant risks in domains where accuracy is essential, such as medical diagnosis, educational assistance, and financial advisory services (Tang et al., 2024; Wang et al., 2024). As organizations increasingly rely on LLMs for complex tasks, the consequences of undetected hallucinations can range from misinformation propagation to decision-making failures, making robust hallucination detection an urgent priority for trustworthy AI deployment.

Existing hallucination detection methods have established foundations across diverse paradigms. Uncertainty-based approaches leverage model confidence signals and entropy to identify potentially unreliable outputs (Manakul et al., 2023; Farquhar et al., 2024). Consistency-based methods evaluate factual reliability through cross-generation agreement and semantic coherence (Li et al., 2023). More recently, attention-based interpretability techniques and representational analysis have provided mechanistic insights into when models exhibit knowledge awareness versus hallucination tendencies (Azaria & Mitchell, 2023; Burns et al., 2022). These methods perform well in their evaluation settings and rely primarily on internal model signals for detection decisions.

However, traditional detection approaches face fundamental limitations when deployed in real-world applications. As illustrated in Figure 1, models often generate hallucinations about recent events, specialized domains, or rapidly evolving information that falls outside their training data coverage (Mallen et al., 2022). Additionally, reliance on internal model signals makes these methods vulnerable to distribution shifts and domain-specific biases that can compromise detection reliability. To address these coverage and recency limitations, retrieval-augmented generation (RAG) has emerged as a promising solution that grounds model outputs in external evidence sources (Lewis et al., 2020; Gao et al., 2023). RAG systems dynamically incorporate relevant documents during generation, enabling models to access up-to-date information while providing explicit evidence for factual claims.

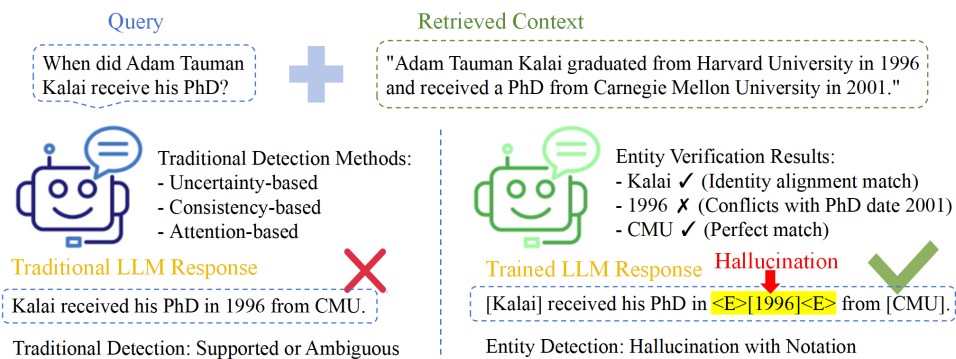

Figure 1: Comparison of traditional and RAG-based entity hallucination detection methods. Example adapted from (OpenAI, 2025).

Despite the promise of evidence-grounded generation, RAG introduces distinct challenges that expose new hallucination detection requirements. Models frequently fabricate entities even when correct information exists within context window, creating misalignments between retrieved evidence and generated content (Niu et al., 2024). Current detection methods typically rely on external judge models for verification, which introduce additional complexity and potential error propagation while operating at token, sentence, or paragraph levels that miss the entity-level factual commitments users most critically evaluate (Yue et al., 2023). This disconnect between detection granularity and user verification behavior motivates a specialized approach to evidence-based hallucination detection.

Addressing these challenges requires overcoming fundamental obstacles that distinguish RAG-based detection from traditional approaches. The spurious correlation problem occurs when hallucinated entities accidentally align with retrieved text through surface-level keyword matches, creating false signals of evidence support. The need for precise localization of factual inconsistencies at the entity level becomes critical, as entities constitute the atomic units of factual information that users prioritize when evaluating response trustworthiness (Thorne et al., 2018). When critical entities such as names, dates, or quantities contain errors, user confidence in the entire response deteriorates regardless of overall semantic coherence. This motivates our central question:

> *How can we leverage RAG to enhance hallucination detection by establishing direct entity-evidence alignment within retrieved contexts?*

Building on this foundation, we propose Evidence-Aligned Entity Verification (EAEV), a novel method that operates entirely within retrieved contexts to verify entity mentions through complementary alignment mechanisms. EAEV evaluates each entity along three dimensions: identity alignment for direct matches, semantic alignment for paraphrases, and consistency alignment for quantitative attributes and conflicts. To address spurious correlations, EAEV incorporates counterfactual stability analysis that distinguishes robust evidence support from fragile alignments. Extensive experiments demonstrate EAEV's effectiveness, achieving 87.89% AUROC on LLaMA2-13B with strong generalization across datasets. Our main contributions are summarized as follows:

- We establish *RAG-based hallucination detection* (RHD) as a novel problem formulation that leverages RAG for entity-level verification within retrieved contexts, tackling the remaining challenge that prior methods rely on internal uncertainty or external judges without evidence traceability.
- We propose *Evidence-Aligned Entity Verification* (EAEV), a novel method that combines multi-dimensional alignment with counterfactual stability analysis to distinguish genuine evidence support from spurious correlations in RAG settings.
- We demonstrate superior performance and generalization across multiple RAG benchmarks and model architectures, achieving state-of-the-art results while maintaining practical deployability.

## 2 PRELIMINARY

In this section, we present necessary notations and establish the theoretical foundation for RAG-based hallucination detection, emphasizing entity-centric verification within retrieved contexts.

**Basic Definitions** Following standard conventions, we represent an LLM as a probability distribution $P_\theta(\cdot)$ over token sequences, where $\theta$ denotes the model parameters. Given a query $q = [x_1, \ldots, x_k]$, the model generates an answer $Y = [x_{k+1}, \ldots, x_{k+l}]$ through autoregressive prediction $P_\theta(x_j | x_1, \ldots, x_{j-1})$. For dataset representation, each instance consists of a query $q$, generated answer $Y$, and retrieved context passages $\mathcal{P} = \{p_k\}_{k=1}^K$. Each answer receives a binary hallucination label $y \in \{0, 1\}$ where $y = 1$ indicates truthful content.

**Traditional Hallucination Detection** Traditional hallucination detection aims to identify factually incorrect content in LLM outputs. Given a query $q$ and answer $Y$, a detector $D$ produces $\hat{y} = D(q, Y)$ where $\hat{y} \in \{0, 1\}$ indicates hallucination presence. Existing methods operate through uncertainty estimation, consistency checking, or external verification, but face challenges when evidence is explicitly available yet underutilized in RAG settings.

**RAG-based Hallucination Detection** RAG-based hallucination detection (RHD) represents a fundamental shift from traditional approaches by leveraging retrieved evidence for verification. Unlike conventional methods that rely solely on model internals, RHD operates under the assumption that factual accuracy can be determined through explicit alignment between generated content and available evidence within retrieved contexts $\mathcal{P}$. We formalize RHD as follows:

Given a query $q$, retrieved contexts $\mathcal{P}$, and generated answer $Y$, the objective of RHD is to learn a detector $D$ that determines factual accuracy through evidence alignment:

$$D(q, Y, \mathcal{P}) = \begin{cases} 1, & \text{if } Y \text{ is supported by evidence in } \mathcal{P}, \\ 0, & \text{otherwise.} \end{cases} \tag{1}$$

The key insight is that factual errors in RAG settings manifest primarily at the entity level, where specific named entities, temporal expressions, and quantities determine overall response reliability.

**Entity-Centric Verification Framework** For entity-centric verification, we extract candidate mentions $s$ from the generated answer $Y$, where each mention has type $t \in \{\text{ENT}, \text{NUM}, \text{NP}\}$ corresponding to named entities, numerical values, and noun phrases. For each mention $s$, we retrieve evidence windows from $\mathcal{P}$ and select primary evidence $e^*$ through relevance scoring. We define three core alignment functions: identity alignment $\text{Id}(s, e^*) \in [0, 1]$ measuring surface correspondence, semantic alignment $\text{Sem}(s, e^*) \in [-1, 1]$ capturing meaning preservation, and consistency alignment $\text{Con}(s, e^*) \in [0, 1]$ evaluating quantitative agreement and conflict detection.

For each mention $s$, we compute support signals through weighted combination of alignment dimensions and detect conflicts through binary indicators. To distinguish robust evidence from spurious correlations, we apply counterfactual stability analysis using perturbation sets $\mathcal{U}$. Finally, mentions are aggregated into entity-level decisions through canonicalization, producing interpretable verification scores with direct evidence traceability.

## 3 METHODOLOGY

### 3.1 MOTIVATION AND OBSERVATIONS

Effective RAG verification requires understanding how factual errors manifest in the presence of relevant evidence. As illustrated in Figure 1, traditional hallucination detection methods rely solely on internal model signals and are limited by training data coverage, while our RAG-enhanced approach incorporates external evidence sources to improve detection accuracy and coverage. Consider a model given documents stating "Adam Tauman Kalai graduated from Harvard University in 1996 and received a PhD from Carnegie Mellon University in 2001" but generating "Kalai received his PhD in 1996 from CMU" (OpenAI, 2025). This example illustrates a fundamental challenge: models can fabricate specific entities while correctly incorporating other factual elements from the context, as noted by recent analysis of why language models hallucinate.

Existing detection methods operating at sentence or paragraph levels fail to localize such precise factual inconsistencies, as the overall semantic coherence remains high despite the critical entity-level error. Empirical analysis across RAG benchmarks reveals that entity-level inconsistencies constitute

the primary failure mode, with named entities, dates, and quantities representing the most frequent error types that directly impact user trust and system reliability. This observation motivates our entity-centric approach: rather than evaluating global semantic consistency, we decompose verification into atomic factual units where evidence alignment can be precisely established and traced.

## 3.2 FRAMEWORK OVERVIEW

To address these challenges, we propose EAEV, which transforms entity verification into a systematic evidence alignment task through four interconnected stages that maintain evidence traceability throughout verification. As shown in Figure 2, the framework operates under three core principles: context-only verification where all signals derive from alignment between generated content and retrieved evidence, entity-centric aggregation enabling cross-mention evidence consolidation, and unified verification architecture supporting both rule-based decisions and model fine-tuning.

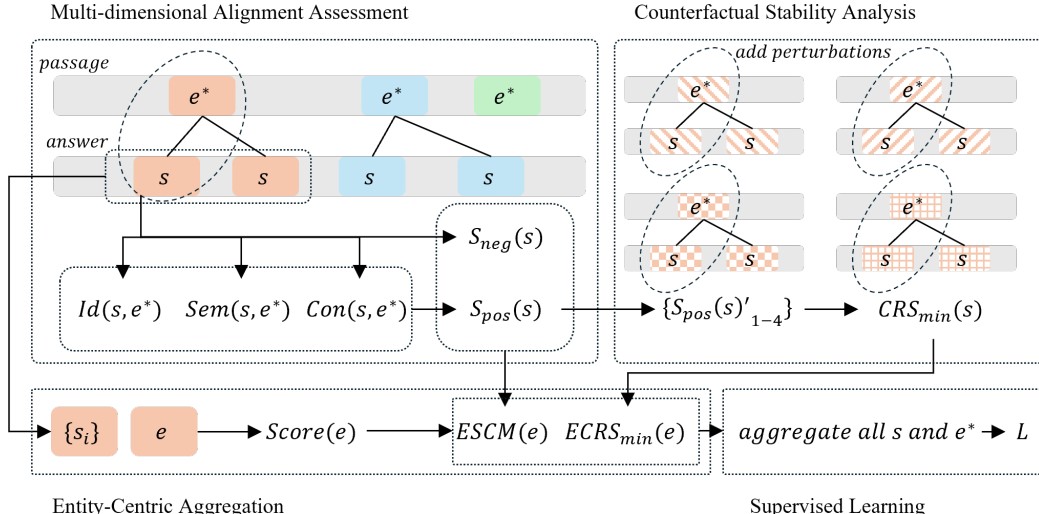

Figure 2: Framework Overview of our Methodology.

Given a query $q$, retrieved context $\mathcal{P}$, and generated answer $Y$, EAEV proceeds through: (1) *candidate mention extraction* that identifies factual commitments in $Y$ and constructs local answer windows, (2) *evidence retrieval and selection* that retrieves top-k evidence windows from $\mathcal{P}$ and selects primary evidence $e^*$ by maximizing relevance scores combining identity and semantic similarity, (3) *multi-dimensional alignment assessment* that evaluates entity-evidence correspondence through complementary signals while testing robustness via stability analysis, and (4) *entity-centric aggregation* that consolidates mention-level signals into entity-level decisions and produces span, entity, and answer-level verification outputs. This modular design enables comprehensive verification while preserving evidence traceability for interpretable decisions.

## 3.3 MULTI-DIMENSIONAL ALIGNMENT ASSESSMENT

For each candidate mention $s$ extracted from the answer and its selected primary evidence $e^*$ from the retrieved context, we evaluate alignment through three complementary dimensions that capture orthogonal aspects of evidential support.

**Identity Alignment** Identity alignment captures direct matches through lexical forms and aliases, providing precise signals for exact alignment. This dimension uses a normalized similarity function that blends exact substring matching with fuzzy token-level matching:

$$\text{Id}(s, e^*) = \max\left(\mathbb{I}[s \subseteq e^* \vee e^* \subseteq s], \text{TSR}(s, e^*)\right) \tag{2}$$

where $\mathbb{I}[\cdot]$ is the indicator function and $\text{TSR}(s, e^*) \in [0, 1]$ computes the normalized token set ratio measuring lexical overlap between mention and evidence tokens. This formulation prioritizes exact matches while gracefully handling orthographic variations and aliases through the fuzzy matching fallback, ensuring robust identity detection across diverse lexical forms.

**Semantic Alignment**   Semantic alignment evaluates meaning preservation through embedding similarity, capturing paraphrases and reformulations that maintain factual content despite variations:

$$\text{Sem}(s, e^*) = \cos(f_{\text{enc}}(s), f_{\text{enc}}(e^*)) \tag{3}$$

where $f_{\text{enc}}(\cdot)$ represents sentence-level embedding encoding that captures semantic correspondence beyond explicit textual correspondence. This approach enables detection of semantically equivalent expressions while maintaining computational efficiency, though it requires careful calibration to prevent accepting spurious semantic matches that lack genuine factual grounding.

**Consistency Alignment**   Consistency alignment addresses value correspondence and explicit factual conflicts through numerical overlap assessment combined with rule-based contradiction detection. For entities with quantitative attributes, we measure consistency through normalized intersection over union of extracted numerical values:

$$\text{Con}(s, e^*) = \frac{|N(s) \cap N(e^*)|}{|N(s) \cup N(e^*)|} + b_{\text{anc}} \cdot \mathbb{I}[\text{anchor}(q, e^*)] \tag{4}$$

where $N(\cdot)$ extracts and normalizes numerical values from text, and $b_{\text{anc}}$ provides an anchor bonus when evidence contains key terms from the original query, strengthening confidence in relevant retrievals. Additionally, we detect explicit contradictions through $S_{\text{neg}}(s) \in \{0, 1\}$ using rule-based patterns that identify temporal mismatches, numerical conflicts, and relational inconsistencies, providing high-precision negative signals that complement the positive consistency scores.

**Type-Adaptive Support Synthesis**   We synthesize the three alignment dimensions into a unified support score that adapts to mention types, recognizing that different entity categories require different verification emphases. For each mention $s$ of type $t \in \{\text{ENT}, \text{NUM}, \text{NP}\}$, we compute positive support signals $S_{\text{pos}}(s) \in [0, 1]$ through weighted combination of alignment dimensions:

$$S_{\text{pos}}(s) = w_I^{(t)} \cdot \text{Id}(s, e^*) + w_S^{(t)} \cdot \text{Sem}(s, e^*) + w_C^{(t)} \cdot \text{Con}(s, e^*) \tag{5}$$

where type-adaptive weights $w_I^{(t)}, w_S^{(t)}, w_C^{(t)}$ are optimized for each mention type—numerical mentions emphasize consistency alignment while named entities prioritize identity and semantic alignment. Additionally, we detect explicit conflicts through binary indicators $S_{\text{neg}}(s) \in \{0, 1\}$ using rule-based patterns that identify temporal mismatches, numerical conflicts, and relational inconsistencies. We then compute a consistency margin $\text{SCM}(s) = S_{\text{pos}}(s) - \beta \cdot S_{\text{neg}}(s)$ where $\beta$ controls conflict penalties, integrating positive evidence support with negative conflict signals.

### 3.4 COUNTERFACTUAL STABILITY ANALYSIS

A critical challenge in RAG-based verification is distinguishing genuine evidence support from spurious correlations, where hallucinated content accidentally matches retrieved text through surface-level similarities. Traditional alignment metrics can be deceived by coincidental keyword overlaps or formatting artifacts that create false signals of factual support. To address this fundamental problem, we propose counterfactual stability analysis that tests whether evidence alignment remains robust under controlled perturbations.

The core insight is that genuine factual correspondence should persist across minor variations in text presentation, while spurious matches are inherently fragile and collapse when surface features change. We define perturbation sets $\mathcal{U}$ containing controlled variations that preserve semantic content while altering surface characteristics. For each mention $s$, we compute stability bounds: minimum support $\text{CRS}_{\min}(s) = \min_{u \in \mathcal{U}} S_{\text{pos}}^{(u)}(s)$ measuring the lowest support under perturbations, and stability gaps $\text{CRS}_{\Delta}(s) = S_{\text{pos}}(s) - \text{CRS}_{\min}(s)$ indicating robustness to variations.

We instantiate $\mathcal{U}$ with four targeted perturbations that address distinct sources of spurious correlation: (1) *leave-one-out evidence removal* eliminates the strongest evidence window to test dependency on single sources, preventing over-reliance on potentially misleading context; (2) *punctuation and case normalization* removes formatting artifacts and capitalization patterns creating false lexical matches, ensuring alignment reflects genuine content rather than presentation; (3) *whitespace compression* eliminates spacing variations and tokenization inconsistencies that might artificially inflate similarity scores; and (4) *alphanumeric-only filtering* retains only core semantic content by removing symbols and special characters that could create spurious token-level alignments.

High minimum support $\text{CRS}_{\min}(s)$ indicates that evidence alignment persists across these controlled variations, indicating the factual correspondence is robust rather than circumstantial. Conversely, large stability gaps $\text{CRS}_{\Delta}(s)$ reveal fragile correlations that depend on specific textual configurations, flagging potentially unreliable evidence support. This stability analysis enables EAEV to distinguish authentic factual grounding from accidental surface-level matches, significantly improving detection precision in challenging cases where traditional alignment metrics alone prove insufficient.

### 3.5 Entity-Centric Aggregation

For entities with multiple mentions across the answer, we consolidate evidence signals to obtain robust entity-level assessments. We canonicalize mention strings through lowercasing, punctuation and article removal to identify coreferent mentions, and apply lightweight pronoun resolution that links pronouns to the most recent non-pronoun entity.

For an entity $e$ with mention set $\{s_i\}$, we aggregate verification signals conservatively. Positive support uses top-K averaging $\text{ES}_{\text{pos}}(e) = \text{mean}(\text{topK}\{S_{\text{pos}}(s_i)\})$ to emphasize strongest evidence across mentions. Negative signals use max pooling $\text{ES}_{\text{neg}}(e) = \max\{S_{\text{neg}}(s_i)\}$ for conservative conflict detection, ensuring any mention-level conflict propagates to entity level. Stability becomes $\text{ECRS}_{\min}(e) = \min\{\text{CRS}_{\min}(s_i)\}$ to identify the weakest link across all entity mentions.

The entity consistency margin $\text{ESCM}(e) = \text{ES}_{\text{pos}}(e) - \beta_e \cdot \text{ES}_{\text{neg}}(e)$ integrates these consolidated signals, where $\beta_e$ controls entity-level conflict penalties. The final entity verification score integrates consistency and stability through a multiplicative combination:

$$\text{score}(e) = \sigma(-\text{ESCM}(e)) \cdot (1 - \sigma(\text{ECRS}_{\min}(e))) \tag{6}$$

where $\sigma(\cdot)$ denotes the sigmoid function. This formulation produces high risk scores for entities with weak evidence support or low stability, enabling answer-level assessment through max pooling over entity scores while preserving traceability to specific evidence windows.

### 3.6 EAEV-Guided Supervised Learning

The alignment and stability signals computed by EAEV provide direct supervision for training models to perform interpretable hallucination detection through entity-level annotation. Rather than requiring complex architectural modifications, we leverage EAEV's verification capabilities to construct high-quality training data where models learn to reproduce answers while marking unsupported entities with verification tags.

For each training instance, we generate target sequences where entities with $\text{ESCM}(e) < \tau_{\text{threshold}}$ are enclosed in $\langle\text{E}\rangle$ markers, creating supervision that directly transfers EAEV's multi-dimensional verification logic to generation. We optimize a token-weighted cross-entropy that transfers EAEV's entity-level signals into generation:

$$L = \sum_t w_t \cdot \text{CE}(p_\theta(y_t|x, y_{<t}), y_t) \tag{7}$$

where

$$w_t = \text{clip}\left(1 + \alpha \cdot \max_{e \ni t} \sigma(-\text{ESCM}(e)) + \gamma \cdot \max_{e \ni t}(1 - \sigma(\text{ECRS}_{\min}(e))), w_{\min}, w_{\max}\right) \tag{8}$$

and tokens outside any tagged entity use $\max_{e \ni t} = 0$. This approach enables standard supervised fine-tuning to learn EAEV's sophisticated verification patterns, transferring interpretable entity-level detection capabilities into generation without requiring specialized decoding procedures or multi-model coordination.

## 4 Experiment

### 4.1 Experiment Settings

**Datasets** We evaluate EAEV across three representative RAG hallucination benchmarks that cover diverse reasoning scenarios and evaluation granularities. **RAGTruth** provides high-quality manual

annotations with nearly 18,000 responses from multiple LLMs across question answering, data-to-text generation, and news summarization tasks, offering fine-grained word-level annotations that we aggregate to answer-level evaluation (Niu et al., 2024). **HotpotQA** represents multi-hop reasoning challenges built on Wikipedia articles with sentence-level supporting facts, requiring cross-document evidence synthesis for accurate verification (Yang et al., 2018). **DelucionQA** focuses on domain-specific hallucinations in automotive manuals with human-annotated labels, providing specialized evaluation for technical content verification (Sadat et al., 2023). More details of the datasets could be found in Appendix. This dataset combination ensures comprehensive evaluation across commonsense reasoning, knowledge-intensive tasks, and domain-specific applications while maintaining consistency in answer-level hallucination assessment.

**Baselines** We conduct experiments on three representative LLMs: Qwen2.5-7B (Yang et al., 2024), LLaMA2-7B (Touvron et al., 2023), and LLaMA2-13B (Touvron et al., 2023). For hallucination detection methods, we compare against eleven state-of-the-art baselines spanning different detection paradigms. Uncertainty-based approaches include **SelfCheckGPT** (Manakul et al., 2023), **Semantic Entropy** (Kuhn et al., 2023), and **LLM-Check** (Jain et al., 2024), which leverage model confidence signals and internal activations for detection. Consistency-based methods such as **Early-Detect** (Snyder et al., 2024) and **NoVo** (Ho et al., 2024) evaluate reliability through cross-generation agreement and attention-level analysis. RAG-specific approaches include **RAGAS** (Es et al., 2024), **RefChecker** (Hu et al., 2024), **ReDEeP** (Sun et al., 2024), and **TSV** (Park et al., 2025), which explicitly incorporate retrieved evidence for verification. We also include general detection methods **Linear Probe** (Duan et al., 2024) and **HaloScope** (Du et al., 2024) that operate on model representations. We evaluate all methods using AUROC, Accuracy, and F1 score as our primary metrics to ensure comprehensive performance assessment. Detailed baseline configurations and implementation details are provided in the Appendix B and A.1.2.

## 4.2 Experimental Results and Analysis

**Main Results** EAEV achieves consistent superiority across all evaluation settings, demonstrating the effectiveness of entity-centric evidence alignment for RAG hallucination detection. As shown in Table 1, our method attains 85.93% average AUROC on Qwen2.5-7B, 84.25% on LLaMA2-7B, and 87.55% on LLaMA2-13B, representing substantial improvements of 2.80, 2.40, and 3.34 percentage points respectively over the strongest baseline TSV. The robustness of performance across diverse model architectures validates our core hypothesis that entity-level factual errors constitute a model-agnostic challenge in RAG systems. Notably, larger models show particularly pronounced improvements, with LLaMA2-13B achieving the highest absolute performance, suggesting that EAEV effectively leverages enhanced model capabilities for more sophisticated evidence alignment while maintaining consistent gains across different architectural families.

Cross-dataset evaluation reveals EAEV's strong generalization capabilities across diverse reasoning scenarios and domain requirements. On RAGTruth's fine-grained annotations, our method achieves the most substantial improvements, demonstrating effectiveness in detecting nuanced factual inconsistencies within general knowledge contexts. HotpotQA results highlight EAEV's strength in multi-hop reasoning scenarios, where our consistency alignment mechanism proves particularly valuable for verifying complex logical chains that span multiple evidence sources. DelucionQA performance validates applicability to specialized technical domains, where entity verification demands precision in handling domain-specific terminology and quantitative relationships. This consistent performance across datasets with fundamentally different characteristics—from general knowledge to multi-hop reasoning to technical domains—confirms that our multi-dimensional alignment framework captures universal patterns in entity-level hallucination detection rather than dataset-specific artifacts.

The performance scaling pattern provides additional validation of our design principles while highlighting EAEV's practical deployability across diverse computational environments. Both 7B and 13B model variants benefit significantly from our approach, with the scaling behavior indicating that our framework harnesses enhanced model capabilities without sacrificing robustness at smaller scales. This versatility enables deployment in resource-constrained settings requiring smaller models while maximizing performance in high-capacity applications. The consistent benefits across all tested architectures, combined with our method's ability to achieve state-of-the-art results through

Table 1: Performance comparison across different models and datasets. We report AUROC, Accuracy (Acc), and F1 scores for each method on three datasets. All results are averaged over three independent runs, with the rightmost columns showing metrics averaged across datasets.

| Method | RAGTruth | | | HotpotQA | | | DelucionQA | | | Average | | |
|---|---|---|---|---|---|---|---|---|---|---|---|---|
| | AUROC | Acc | F1 | AUROC | Acc | F1 | AUROC | Acc | F1 | AUROC | Avg Acc | Avg F1 |
| *Qwen2.5-7B* | | | | | | | | | | | | |
| EarlyDetect | 66.38 | 70.12 | 65.34 | 67.15 | 69.32 | 65.10 | 68.25 | 69.83 | 66.21 | 67.24 | 69.72 | 65.55 |
| Selfcheckgpt | 64.39 | 69.05 | 63.28 | 65.73 | 68.32 | 63.95 | 66.43 | 68.76 | 65.01 | 65.51 | 68.71 | 64.08 |
| Novo | 73.79 | 76.21 | 68.67 | 74.85 | 75.75 | 69.12 | 76.03 | 76.12 | 70.43 | 74.89 | 76.01 | 69.44 |
| Linear Probe | 75.27 | 77.38 | 69.52 | 76.11 | 77.02 | 69.84 | 77.10 | 77.54 | 71.02 | 76.16 | 77.31 | 70.13 |
| HaloScope | 71.01 | 74.16 | 67.70 | 72.24 | 73.90 | 68.20 | 73.01 | 74.16 | 69.05 | 72.08 | 74.05 | 68.32 |
| LLM-Check | 62.75 | 68.07 | 62.18 | 63.91 | 67.20 | 62.93 | 65.14 | 67.86 | 64.23 | 63.93 | 67.71 | 63.09 |
| Semantic Entropy | 65.43 | 70.65 | 64.71 | 66.87 | 69.83 | 65.25 | 68.02 | 70.19 | 65.84 | 66.75 | 70.19 | 65.27 |
| RAGAS | 74.76 | 77.32 | 69.90 | 76.02 | 76.84 | 70.40 | 76.89 | 77.01 | 71.43 | 75.89 | 77.06 | 70.58 |
| RefCheck | 73.25 | 75.89 | 68.43 | 74.61 | 75.30 | 68.81 | 75.20 | 75.76 | 70.01 | 74.35 | 75.65 | 69.08 |
| ReDEeP | 77.87 | 78.51 | 71.92 | 79.43 | 78.23 | 72.74 | 80.12 | 78.96 | 73.52 | 79.14 | 78.57 | 72.71 |
| TSV | 81.45 | 79.83 | 72.37 | 82.07 | 79.36 | 72.28 | 85.87 | 80.47 | 74.97 | 83.13 | 79.89 | 73.21 |
| **EAEV (Ours)** | **85.36** | **80.04** | **74.28** | **86.74** | **81.23** | **74.35** | 85.68 | 80.25 | **75.62** | **85.93** | **80.51** | **74.75** |
| *LLaMA2-7B* | | | | | | | | | | | | |
| EarlyDetect | 65.12 | 68.87 | 63.98 | 66.23 | 68.05 | 63.55 | 67.12 | 68.42 | 64.33 | 66.16 | 68.42 | 63.95 |
| Selfcheckgpt | 63.18 | 67.31 | 62.01 | 64.45 | 66.25 | 62.74 | 65.37 | 66.90 | 63.45 | 64.33 | 66.82 | 62.72 |
| Novo | 72.25 | 75.28 | 67.12 | 73.31 | 74.82 | 67.62 | 74.38 | 75.01 | 67.89 | 73.31 | 75.03 | 67.89 |
| Linear Probe | 73.56 | 76.36 | 68.23 | 74.25 | 75.43 | 68.55 | 75.48 | 75.62 | 70.02 | 74.44 | 75.68 | 68.92 |
| HaloScope | 69.83 | 73.24 | 66.31 | 70.92 | 72.43 | 66.75 | 71.74 | 73.25 | 67.45 | 70.83 | 72.96 | 66.82 |
| LLM-Check | 61.47 | 66.42 | 60.08 | 62.63 | 65.32 | 61.32 | 63.71 | 66.19 | 62.03 | 62.59 | 65.95 | 61.34 |
| Semantic Entropy | 64.12 | 69.01 | 63.43 | 65.25 | 68.32 | 63.78 | 66.30 | 69.02 | 64.01 | 65.22 | 68.78 | 63.71 |
| RAGAS | 73.11 | 76.25 | 68.11 | 74.43 | 75.62 | 68.75 | 75.45 | 76.12 | 70.12 | 74.33 | 76.69 | 68.99 |
| RefCheck | 71.66 | 74.83 | 66.92 | 73.08 | 74.41 | 67.30 | 74.02 | 74.88 | 68.15 | 72.92 | 74.71 | 67.46 |
| ReDEeP | 76.42 | 78.01 | 71.23 | 77.63 | 77.15 | 71.94 | 78.35 | 77.66 | 72.43 | 77.47 | 77.61 | 71.86 |
| TSV | 82.04 | 79.12 | 72.01 | 82.64 | 78.83 | 72.10 | 80.88 | 77.52 | 73.95 | 81.85 | 78.49 | 72.77 |
| **EAEV (Ours)** | **84.57** | **80.12** | **74.07** | **84.96** | **82.55** | **73.39** | **83.21** | **78.32** | **74.45** | **84.25** | **80.33** | **73.97** |
| *LLaMA2-13B* | | | | | | | | | | | | |
| EarlyDetect | 67.18 | 70.01 | 65.12 | 68.42 | 69.05 | 65.87 | 69.66 | 69.81 | 66.50 | 68.42 | 69.62 | 65.83 |
| Selfcheckgpt | 65.47 | 68.10 | 63.02 | 66.93 | 67.22 | 63.89 | 67.82 | 67.88 | 64.52 | 66.74 | 67.73 | 63.81 |
| Novo | 74.81 | 76.35 | 69.31 | 75.84 | 76.11 | 70.16 | 77.12 | 76.55 | 71.22 | 75.92 | 76.34 | 70.21 |
| Linear Probe | 76.31 | 77.65 | 70.54 | 77.54 | 77.33 | 71.39 | 78.82 | 77.97 | 72.22 | 77.55 | 77.63 | 71.53 |
| HaloScope | 71.74 | 74.45 | 68.20 | 72.81 | 73.25 | 68.91 | 73.93 | 74.15 | 69.85 | 72.83 | 73.93 | 68.99 |
| LLM-Check | 63.83 | 67.12 | 61.95 | 65.28 | 66.52 | 62.73 | 66.55 | 66.98 | 63.45 | 65.19 | 66.87 | 62.71 |
| Semantic Entropy | 66.02 | 70.20 | 64.62 | 67.35 | 69.02 | 65.21 | 68.40 | 70.10 | 65.83 | 67.26 | 69.77 | 65.22 |
| RAGAS | 75.67 | 77.32 | 70.22 | 76.88 | 77.01 | 71.01 | 78.02 | 77.66 | 72.52 | 76.86 | 77.33 | 71.25 |
| RefCheck | 74.25 | 76.00 | 68.85 | 75.66 | 75.43 | 69.33 | 76.92 | 76.41 | 70.44 | 75.61 | 75.95 | 69.54 |
| ReDEeP | 78.93 | 79.63 | 72.33 | 80.11 | 79.25 | 73.21 | 81.04 | 79.81 | 74.15 | 80.03 | 79.56 | 73.23 |
| TSV | 84.55 | 80.12 | 73.50 | 83.12 | 79.43 | 73.22 | 84.96 | 80.12 | 75.68 | 84.21 | 79.89 | 74.13 |
| **EAEV (Ours)** | **87.89** | **84.29** | **76.85** | **88.12** | **83.53** | **75.59** | **86.65** | **83.22** | **77.92** | **87.55** | **83.68** | **76.79** |

entity-centric verification within retrieved contexts, establishes EAEV as a reliable and scalable solution for RAG hallucination detection across varied deployment scenarios.

**Ablation Study** To validate each component's contribution, we conduct comprehensive ablation studies across all benchmarks. As shown in Figure 3a and Table 2, counterfactual stability analysis provides the most substantial contribution, confirming the necessity of our approach for distinguishing genuine evidence support from spurious correlations. The results demonstrate that each alignment dimension contributes meaningfully to overall performance, with balanced degradation patterns indicating that all components address distinct verification challenges. The full framework's superior performance validates our multi-dimensional design philosophy and demonstrates synergistic effects among complementary alignment mechanisms. Details are provided in Appendix C.1.

**Sensitivity Analysis** We analyze EAEV's robustness to answer-side window length, a key parameter controlling contextual span during evidence alignment. As shown in Figure 3b, the framework achieves optimal performance with 30-token windows while maintaining stability across the practical range. Smaller windows limit contextual information for accurate alignment, while larger windows introduce noise that dilutes alignment signals. The framework demonstrates reasonable robustness within the 25-35 token range, validating our parameter choice and confirming consistent performance across deployment scenarios. This analysis establishes EAEV's reliability and practical applicability under varying configuration settings. Details are provided in Appendix C.2.

Table 2: Ablation study on LLaMA2-13B model. Results are reported as AUROC, Accuracy (Acc), and F1 on three benchmarks. The rightmost columns show averaged metrics across all datasets.

| Variant | RAGTruth | | | HotpotQA | | | DelucionQA | | | Average | | |
|---|---|---|---|---|---|---|---|---|---|---|---|---|
| | AUROC | Acc | F1 | AUROC | Acc | F1 | AUROC | Acc | F1 | AUROC | Acc | F1 |
| w/o Identity | 85.10 | 82.00 | 74.70 | 86.00 | 81.30 | 73.20 | 84.20 | 81.00 | 75.80 | 85.10 | 81.43 | 74.57 |
| w/o Semantic | 83.80 | 81.40 | 73.60 | 85.40 | 81.10 | 73.10 | 83.00 | 80.80 | 75.60 | 84.07 | 81.10 | 74.10 |
| w/o Consistency | 85.60 | 82.10 | 74.50 | 83.40 | 79.30 | 71.00 | 81.80 | 78.90 | 73.30 | 83.60 | 80.10 | 72.93 |
| w/o Stability | 82.10 | 79.80 | 72.40 | 82.30 | 79.00 | 71.60 | 81.70 | 78.70 | 73.00 | 82.03 | 79.17 | 72.33 |
| Full | 87.89 | 84.29 | 76.85 | 88.12 | 83.53 | 75.59 | 86.65 | 83.22 | 77.92 | 87.55 | 83.68 | 76.79 |

(a) Ablation Study on Different Components

(b) Window Size Sensitivity Analysis

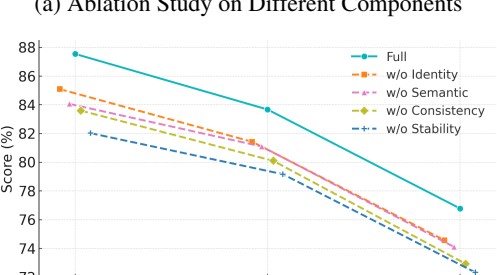
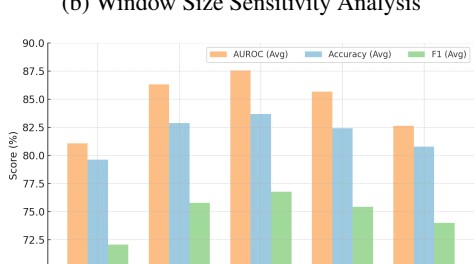

Figure 3: Ablation study results on LLaMA2-13B, showing ablation analysis on different components (left) and window size sensitivity (right).

## 5 RELATED WORK

**Traditional Hallucination Detection**    Traditional hallucination detection methods primarily leverage uncertainty estimation and self-consistency mechanisms within model outputs. Representative approaches include SelfCheckGPT, which measures semantic consistency across multiple generations (Manakul et al., 2023), and Semantic Entropy, which operates on meaning-level divergences (Farquhar et al., 2024). Recent advances explore attention-level interpretability through NoVo (Ho et al., 2025) and representational analysis of knowledge-awareness directions (Ferrando et al., 2025). While effective in controlled settings, these methods remain constrained by their reliance on internal model signals rather than explicit evidence verification.

**Evidence-Based Hallucination Detection in RAG**    RAG environments present unique challenges where hallucinations persist despite available evidence, motivating specialized detection approaches. RARR employs research and revision stages for evidence attribution and consistency-based correction (Gao et al., 2022). FActScore provides atomic-level factual evaluation by decomposing generated text into verifiable claims (Min et al., 2023). CoVe introduces systematic self-verification through question generation and independent answering (Dhuliawala et al., 2024). ReDeEP leverages mechanistic interpretability to disentangle parametric and contextual knowledge contributions (Sun et al., 2024), while RAGTruth establishes evaluation infrastructure with fine-grained annotations (Niu et al., 2024). These approaches highlight the importance of evidence-grounded verification but typically operate at coarse granularities or require external verification mechanisms. Our work addresses this limitation through entity-level verification within retrieved contexts, providing direct evidence traceability without dependencies on external judges.

## 6 CONCLUSION

Hallucination detection remains critical for reliable RAG system deployment in factual applications. We introduced EAEV, a novel framework that performs entity-level verification through multi-dimensional evidence alignment and counterfactual stability analysis. By distinguishing genuine factual support from spurious correlations, EAEV addresses fundamental challenges in RAG-based verification where hallucinated content accidentally matches retrieved text. Experimental results demonstrate substantial improvements across benchmarks and model architectures, achieving 87.55% average AUROC on LLaMA2-13B. The framework's strong generalization capabilities and practical deployability establish robust entity-level verification as a highly reliable approach for accurate hallucination detection in modern evidence-grounded generation systems.

ETHICS STATEMENT

Our study adheres to the ICLR Code of Ethics. All experiments were conducted on publicly available datasets and open-source language models, as listed in Appendix A.1. No private, sensitive, or personally identifiable information is involved. The primary objective of this work is to advance the understanding of hallucination detection in large language models, with an emphasis on transparency, fairness, and responsible research practices.

REPRODUCIBILITY STATEMENT

All models and benchmark datasets employed in this study are publicly available. Detailed descriptions of the datasets are given in Appendix A.1.1, while the implementation details of our method are provided in Appendix A.1.2. To ensure reproducibility, all experiments were conducted on four NVIDIA A100 GPUs within a controlled environment, using Python 3.10.18 and PyTorch 2.2.2 (CUDA 11.8).

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

# A  APPENDIX

## A.1  EXPERIMENTAL DETAILS

### A.1.1  DATASETS DETAILS

**RAGTruth** RAGTruth provides a controlled environment for analyzing hallucinations in standard RAG pipelines. The corpus aggregates responses from both open-source and closed-source LLMs, accompanied by meticulous word-level manual annotations and instance-level labels across three task categories: question answering, data-to-text generation, and news summarization. The dataset comprises approximately 18,000 annotated responses in total. We compute all evaluation metrics at the answer level to maintain consistency across comparisons (Niu et al., 2024).

**RAGBench (HotpotQA & DelucionQA)**   RAGBench is a large-scale benchmark containing approximately 100,000 examples with a standardized RAG schema that provides retrieved contexts and answer annotations suitable for hallucination detection tasks. The benchmark spans five domains and twelve component datasets. We utilize two representative components: (i) **HotpotQA**, a multi-hop question answering benchmark built on Wikipedia articles with sentence-level supporting facts that emphasizes cross-document reasoning capabilities; and (ii) **DelucionQA**, a domain-specific QA dataset constructed from automotive user manuals, featuring human-annotated labels that indicate whether answers contain hallucinations given the retrieved context. We adopt the benchmark's evaluation protocol and consistently assess performance at the answer level (Friel et al., 2025; Yang et al., 2018; Sadat et al., 2023).

### A.1.2 IMPLEMENTATION DETAILS

We run all experiments on servers equipped with 4×NVIDIA A100 GPUs and server-grade multi-core processors. Our implementation is based on PyTorch and Hugging Face Transformers. We use LLaMA-Factory for LLM fine-tuning and inference (with LoRA). Unless otherwise specified, we employ greedy search for generation decoding, and all other parameters follow the default settings of each model.

For candidate construction and evidence retrieval, we retain at most 5 candidate windows per mention with $\text{top\_bm25} = 2$ and $\text{top\_embed} = 2$. Answer-side windows use $\text{window\_tokens} = 30$ and $\text{stride} = 15$. For multi-dimensional alignment, we use type-adaptive weights with defaults ENT: $(0.45, 0.45, 0.10)$, NUM: $(0.25, 0.25, 0.50)$, and NP: $(0.35, 0.35, 0.30)$. We set default $\beta = 1.0$ for consistency margin. For CRS analysis, we apply four perturbation types: leave-one-out, depunctuating and lowercasing, compressing whitespace, and retaining only alphanumeric characters. For entity grouping, we use conservative aggregation with default $K = 2$ and $\beta_e = 1.0$. For decision rules, we scan $\tau_{\text{scm}} \in [-0.5, 0.2]$, $\tau_{\text{escm}} \in [-0.5, -0.1]$, $\tau_{\text{crs\_min}} \in [0.0, 0.5]$, and $K \in \{1, 2, 3\}$ on validation sets.

For EAEV-guided SFT, we insert $\langle E \rangle ... \langle /E \rangle$ markers and select $(\alpha, \gamma, w_{\min}, w_{\max})$ on validation sets. Fine-tuning uses LLaMA-Factory with LoRA, following framework defaults except for token weighting and data annotation. All hyperparameters use validation set selection, and final results report best validation configurations.

### A.1.3 EVALUATION METRICS

Following prior works (Kuhn et al., 2023; Du et al., 2024), we employ three complementary metrics to evaluate hallucination detection performance: area under the receiver operating characteristic curve (AUROC), Accuracy, and F1 score.

**AUROC** measures the ability of a method to discriminate between truthful and hallucinated outputs across different decision thresholds. A higher AUROC indicates better overall ranking performance independent of a specific threshold.

**Accuracy** is calculated by comparing predicted labels with ground-truth annotations under a fixed threshold (e.g., 0.5 on the similarity score between the generation and the reference). It reflects the proportion of correctly classified instances but can be biased when classes are imbalanced.

**F1 score**, the harmonic mean of Precision and Recall, provides a balanced evaluation when both false positives and false negatives are costly. It is particularly useful in assessing detection performance under skewed class distributions.

Together, these metrics ensure a comprehensive assessment of both ranking quality and classification reliability in hallucination detection.

### A.1.4 MODEL DETAILS

We conduct our experiments on three widely used large language models that represent different scales and training paradigms. **Qwen2.5-7B** (Yang et al., 2024) is an open-source model from Alibaba's Qwen series, designed with improved pre-training data and instruction tuning for multilingual reasoning. **LLaMA2-7B** (Touvron et al., 2023) and **LLaMA2-13B** (Touvron et al., 2023) are part of Meta's LLaMA2 family, which have been extensively used as backbone models in academic

research and industrial applications. Together, these models cover diverse capacities and training corpora, providing a representative testbed for evaluating hallucination detection methods.

# B  DETAILS ABOUT BASELINE MODELS

We compare our approach against eleven representative hallucination detection baselines. Below we briefly introduce each method and its underlying intuition.

- **SelfCheckGPT** (Manakul et al., 2023): A zero-resource, sampling-based detector that repeatedly queries the model to generate multiple candidate responses and then measures their consistency. Greater inconsistency across samples suggests a higher risk of hallucination, making this method effective even without external evidence.
- **Semantic Entropy** (Kuhn et al., 2023): Estimates hallucination likelihood by computing linguistic invariances in token-level predictive distributions. When semantic alternatives diverge strongly in probability space, the model exhibits higher semantic entropy, indicating uncertainty and potential unreliability in factual grounding.
- **LLM-Check** (Jain et al., 2024): Probes internal hidden states of LLMs with lightweight classifiers to directly flag hallucinations. By exploiting activation-level features, LLM-Check can detect subtle factual errors that do not manifest at the surface level but are encoded within the model's latent representations.
- **Linear Probe** (Duan et al., 2024): A straightforward but effective baseline that trains linear classifiers on the hidden states of LLMs. By mapping internal activations to truthfulness labels, Linear Probe directly tests how much factuality information is encoded within raw model representations.
- **HaloScope** (Du et al., 2024): Leverages large quantities of unlabeled LLM outputs and applies energy-based and representation-driven detectors. By clustering semantic patterns across generations, HaloScope effectively identifies outliers that correspond to hallucinated claims with minimal supervision.
- **EarlyDetect** (Snyder et al., 2024): A proactive detector that monitors generation in-progress. By analyzing partial outputs and their factual signals, EarlyDetect aims to catch hallucinations early, before the model produces fully misleading answers, thus enabling faster correction or intervention.
- **NoVo** (Ho et al., 2024): Stands for Norm Voting off hallucinations. This method measures the norms of attention heads and aggregates their "votes" to infer factual reliability. It leverages attention-level interpretability to highlight internal disagreement patterns that often precede hallucinated generations.
- **RAGAS** (Es et al., 2024): Focuses on retrieval-augmented settings by breaking down model outputs into atomic statements and verifying each against retrieved passages. Faithfulness is quantified as the ratio of supported claims, allowing fine-grained detection of unsupported or fabricated content.
- **RefChecker** (Hu et al., 2024): Constructs structured knowledge graphs from model outputs and checks their alignment with external references. This graph-based perspective enables detection of hallucinations that may not be obvious at sentence level but become evident when relational consistency is examined.
- **ReDEeP** (Sun et al., 2024): Employs mechanistic interpretability in retrieval-augmented generation (RAG). By tracing attention flow from queries to evidence passages, ReDEeP identifies whether the model's factual claims are truly supported by retrieved documents or merely spurious correlations.
- **TSV** (Park et al., 2025): Introduces the Truthfulness Separator Vector, which perturbs latent representations during inference to evaluate the stability of factual claims. Robust claims remain separable under perturbations, while hallucinated ones collapse, offering a novel perspective on truthfulness detection.

Table 3: Ablation study on LLaMA2-13B model. Results are reported as AUROC, Accuracy (Acc), and F1 on three benchmarks. The rightmost columns show averaged metrics across all datasets.

| Variant | RAGTruth | | | HotpotQA | | | DelucionQA | | | Average | | |
|---|---|---|---|---|---|---|---|---|---|---|---|---|
| | AUROC | Acc | F1 | AUROC | Acc | F1 | AUROC | Acc | F1 | AUROC | Acc | F1 |
| w/o Identity | 85.10 | 82.00 | 74.70 | 86.00 | 81.30 | 73.20 | 84.20 | 81.00 | 75.80 | 85.10 | 81.43 | 74.57 |
| w/o Semantic | 83.80 | 81.40 | 73.60 | 85.40 | 81.10 | 73.10 | 83.00 | 80.80 | 75.60 | 84.07 | 81.10 | 74.10 |
| w/o Consistency | 85.60 | 82.10 | 74.50 | 83.40 | 79.30 | 71.00 | 81.80 | 78.90 | 73.30 | 83.60 | 80.10 | 72.93 |
| w/o Stability | 82.10 | 79.80 | 72.40 | 82.30 | 79.00 | 71.60 | 81.70 | 78.70 | 73.00 | 82.03 | 79.17 | 72.33 |
| **Full** | **87.89** | **84.29** | **76.85** | **88.12** | **83.53** | **75.59** | **86.65** | **83.22** | **77.92** | **87.55** | **83.68** | **76.79** |

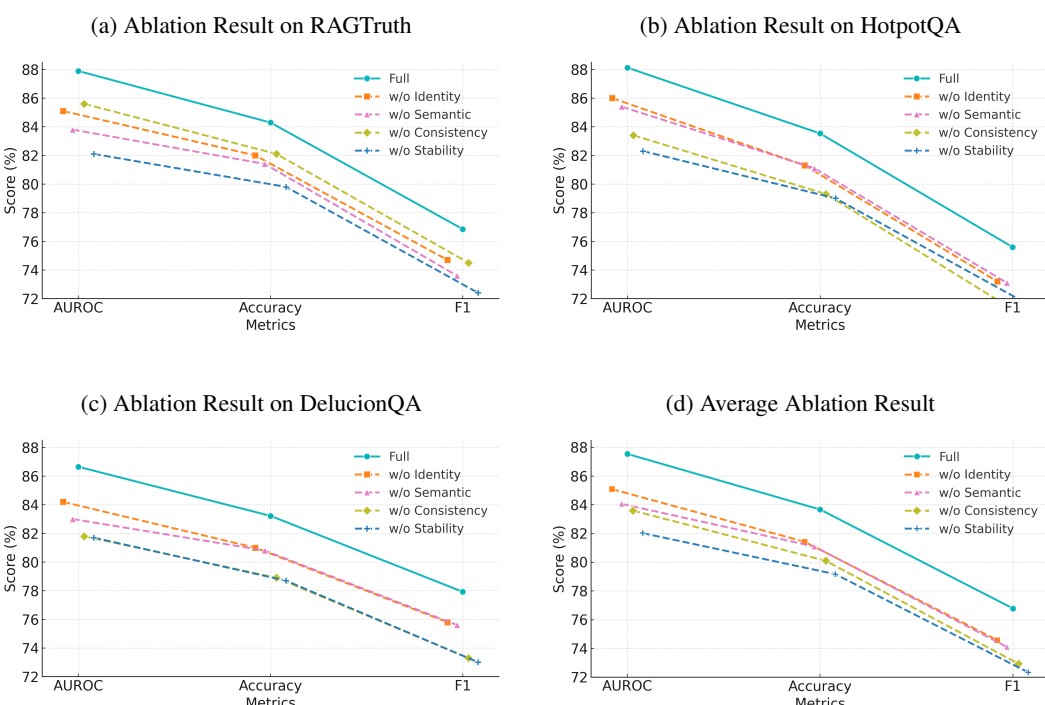

Figure 4: Detailed ablation results on LLaMA2-13B. Results are shown on RAGTruth (top-left), HotpotQA (top-right), DelucionQA (bottom-left), and averaged across datasets (bottom-right).

## C  ADDITIONAL RESULTS

### C.1  COMPLETE ABLATION STUDY

We provide comprehensive ablation analysis across all datasets and model architectures to validate each component's contribution. Table 3 presents the complete ablation results on LLaMA2-13B, while Figures 4a through 4d show detailed performance degradation patterns across individual datasets and averaged results.

The ablation visualizations reveal distinct component contributions across different evaluation scenarios. Counterfactual stability analysis demonstrates the most substantial impact across all datasets, with removal leading to 5.52 AUROC points average degradation. This consistent pattern confirms the necessity of distinguishing genuine evidence support from spurious correlations regardless of dataset characteristics. Consistency alignment shows particularly pronounced effects on HotpotQA and DelucionQA, where quantitative verification becomes critical for multi-hop reasoning and domain-specific content. Semantic alignment exhibits stronger influence on RAGTruth, reflecting its importance for handling paraphrased expressions in general knowledge contexts. Identity alignment provides steady baseline performance through exact matching across all evaluation settings.

Table 4: Sensitivity to answer-side window length (LLaMA2-13B). Results are reported as AUROC, Accuracy (Acc), and F1 on three benchmarks. The rightmost columns show averaged metrics across all datasets.

| window size | RAGTruth | | | HotpotQA | | | DelucionQA | | | Average | | |
|---|---|---|---|---|---|---|---|---|---|---|---|---|
| | AUROC | Acc | F1 | AUROC | Acc | F1 | AUROC | Acc | F1 | AUROC | Avg Acc | Avg F1 |
| 20 | 81.81 | 79.95 | 72.04 | 80.52 | 79.27 | 71.03 | 80.90 | 79.85 | 73.28 | 81.07 | 79.63 | 72.07 |
| 25 | 86.23 | 83.17 | 75.64 | 87.03 | 82.81 | 74.82 | 85.81 | 82.70 | 76.96 | 86.33 | 82.87 | 75.77 |
| **30** | **87.89** | **84.29** | **76.85** | **88.12** | **83.53** | **75.59** | **86.65** | **83.22** | **77.92** | **87.55** | **83.68** | **76.79** |
| 35 | 85.42 | 82.51 | 75.15 | 86.13 | 82.26 | 74.64 | 85.53 | 82.68 | 76.60 | 85.67 | 82.43 | 75.43 |
| 40 | 82.65 | 80.53 | 73.42 | 83.16 | 81.04 | 73.68 | 82.27 | 80.83 | 75.01 | 82.63 | 80.77 | 74.00 |

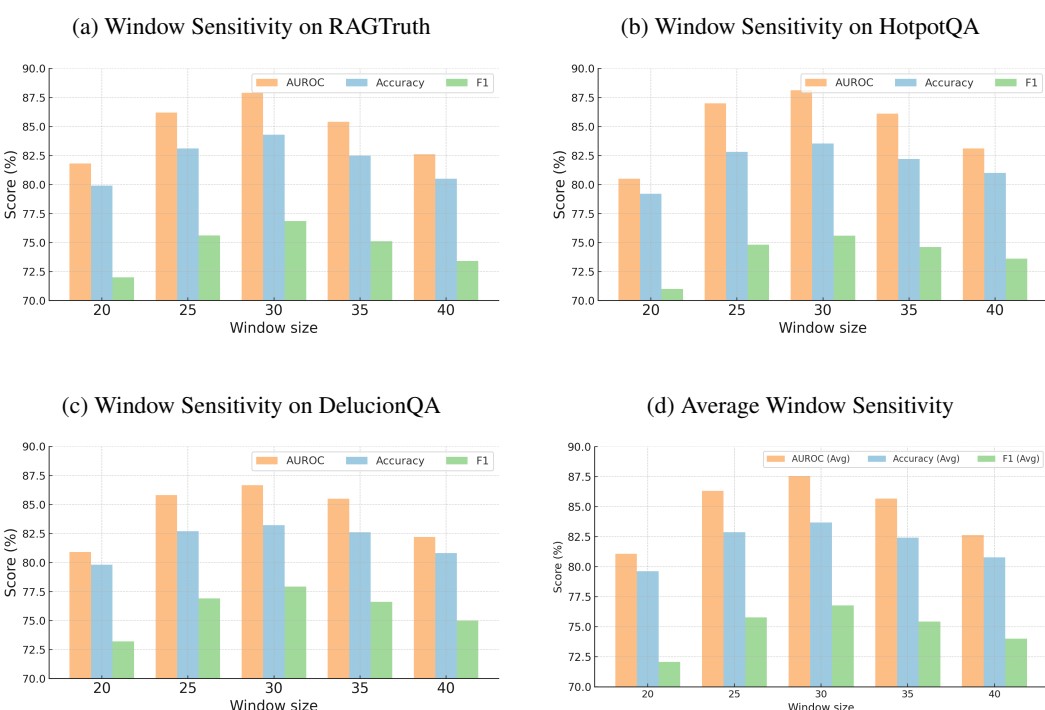

Figure 5: Parameter sensitivity analysis of **EAEV** under different answer-side window lengths. Results are shown on RAGTruth (top-left), HotpotQA (top-right), DelucionQA (bottom-left), and averaged across datasets (bottom-right).

The balanced degradation curves across datasets validate our multi-dimensional design philosophy. Each alignment dimension addresses distinct verification challenges while maintaining complementary effects, with no single component dominating performance. The stability analysis component's consistent importance across all scenarios confirms the practical value of robustness testing in evidence-based verification systems.

## C.2 SENSITIVITY ANALYSIS

Parameter sensitivity analysis demonstrates EAEV's robustness across different configuration settings. Table 4 provides detailed performance under varying answer-side window lengths, while Figures 5a through 5d illustrate the characteristic inverted-U performance curves across individual datasets.

The sensitivity visualizations reveal consistent optimal performance at 30-token windows across all datasets, with graceful degradation patterns for both smaller and larger window sizes. RAGTruth shows the sharpest sensitivity curve, indicating that fine-grained annotations benefit most from optimal contextualization. HotpotQA exhibits broader stability around the optimum, reflecting the method's robustness for multi-hop reasoning tasks. DelucionQA demonstrates intermediate sensi-

---

**Task.** Given a question, supporting passages, and a model answer, mark any unsupported or contradictory entity mentions in the answer using `<E>...</E>` tags. Keep the original answer text and only add tags.

**Question.** {*String*}

**Supporting Passages:**
`<W1>` {*String*} `</W1>`
`<W2>` {*String*} `</W2>`
`<W3>` {*String*} `</W3>`

**Original Answer.** {*String*}

**Instructions.**
- Mark entities contradicted by the supporting passages.
- Mark entities lacking sufficient evidence support.
- Preserve all original text—only add `<E>...</E>` tags.
- Focus on named entities, dates, numbers, and key factual claims.

**Expected Output.** *Answer text with `<E>...</E>` tags around unsupported entities.*

---

Figure 6: Prompt template for EAEV-guided supervised fine-tuning.

tivity patterns, suggesting balanced requirements between contextual information and noise control in domain-specific settings.

Performance degradation below 25 tokens reflects insufficient contextual information for accurate alignment assessment, while degradation above 35 tokens indicates noise introduction from irrelevant content. The framework maintains reasonable stability within the 25-35 token range across all datasets, supporting practical deployment flexibility. These results validate our parameter selection methodology and confirm EAEV's reliability under varying configuration requirements.

## D    DETAILS FOR PROMPT TEMPLATE

The EAEV-guided supervised fine-tuning transforms entity verification into an executable annotation generation task. We design a structured prompt that enables models to learn EAEV's multi-dimensional alignment patterns through standard supervised training while maintaining evidence traceability.

The prompt design follows several key principles to ensure effective knowledge transfer from EAEV's verification framework. The task description explicitly requires minimal modification where only annotation tags are added without altering original answer text. Supporting passages are clearly delineated with window markers (`<W1>`, `<W2>`, `<W3>`) to maintain precise evidence traceability throughout verification. The instructions distinguish between contradicted entities and those lacking sufficient support, reflecting EAEV's multi-dimensional alignment assessment.

This structured approach enables standard supervised fine-tuning to learn sophisticated verification patterns while preserving interpretability through direct evidence grounding. The prompt transforms entity-level hallucination detection into a sequence labeling task that models can learn through token-weighted cross-entropy loss, directly implementing the supervision mechanism described in Section 3.

## E    USAGE CLAIM OF LLMS

We use LLM for grammar and spelling checks only, with prompt "Proofread the sentences". All conceptual development, analysis, writing, and editing were carried out solely by the authors without LLM assistance.

