# OpenReview forum: "Anchoring Entities: Retrieval-Augmented Hallucination Detection"
_ICLR.cc/2026/Conference — ICLR 2026 Conference Withdrawn Submission_

### Official Review · Reviewer_3tmj · 2025-10-14

**Soundness:** 1
**Presentation:** 1
**Contribution:** 2
**Rating:** 2
**Confidence:** 4

**Summary:**

This paper introduced RAG-based hallucination detection, a new task that aims to detect hallucination based on the alignment between LLM-generated text and retrieved documents. The authors then proposed Evidence-Alighed Entity Verification, a new method to detect entity-level hallucination with retrieved documents. The approach consists of alignment assessment, stability analysis, and entity-centric aggregation. The authors conducted experiments on four datasets with three models, showing a better performance against other baselines.

**Strengths:**

1. The authors introduced a novel approach for hallucination detection at the entity level, which will be useful in real-world applications to localize hallucinations
2. The proposed approach achieves the highest performance on almost all datasets and models

**Weaknesses:**

1. **Novelty of RAG-based hallucination detection (RHD).** The author claims that they proposed RHD as a novel task. However, detecting hallucination based on retrieved evidence is not a new task. People have been using NLI models or LLMs to check if the answer contradicts provided evidence [1, 2]. In addition, this idea has also been adopted at the entity level [3].
2. **Unjustified statements.** One major concern of this paper is that it contains many unjustified statements, weakening the sound of this paper. For example, in Line 118, the authors claimed that "[Existing methods] face challenges when evidence is explicitly available yet underutilized in RAG setting" without further explanation or citations. In addition, in Line 132 (and also 161), the authors claimed that "factual errors in RAG settings manifest primarily at the entity level" without providing any citations or empirical evidence in later Sections. As the approach of entity-level RAG-based hallucination detection is the core of this paper, I think these statements should be properly justified.
3. **Lack details of the proposed approach.** Another major concern is that many details of EAEV are missing. For example:
    - Lines 205 and 239: The authors did not explain how they extract $s$. The authors also didn't explain why they only focus on certain types of entity (i.e., ENT, NUM, NP)
    - Line 229: The authors did not explain what $\text{anchor}(q,e^\ast)$ is
    - Line 245: It is unclear what the rule-based patterns are for identifying explicit conflicts
    - Lines 231, 241, and 246: The authors did not explain how they found the hyperparameters
    - The authors should properly justify these design choices or provide experimental/ablation results to support the choices.
4. **Basic assumption.** The basic assumption of the proposed approach is that the evidence retrieved by a RAG system is correct and sufficient to detect hallucination. However, a RAG system can return noise or incorrect data, which may degrade the performance of the proposed approach. The authors should conduct experiments or analyses in such setting, or mention it as a limitation.
5. **Lack entity-level verification.** The authors claim that they proposed an entity-level hallucination detection approach. However, all the experiments are conducted at the answer level. I believe some experiments/analyses at the entity level are necessary to justify their approach.

[1]: Fast and Accurate Factual Inconsistency Detection Over Long Documents. (2023)

[2]: FACTSCORE: Fine-grained Atomic Evaluation of Factual Precision in Long Form Text Generation (2023)

[3]: HalluEntity: Benchmarking and Understanding Entity-Level Hallucination Detection (2025)

**Questions:**

1. What is the purpose of the supervised learning model in the whole pipeline? The authors discussed the supervised model in Sec 3.6 but did not conduct any experiment on it. There are also other details about the model (e.g., what dataset is used to train the model). It is unclear to me why the authors included this section.
2. Fig 3: For the left figure, there is no relationship between AUROC, Accuracy, and F1. Thus, using a line plot to connect these three doesn't make sense to me. The authors should consider using a bar plot instead. On the other hand, it would be better to use a line plot in the right figure.

---

> ### Author Response · Authors · 2025-11-21
> **Author Response to Reviewer Comments**
>
> We thank the reviewer for the careful and detailed feedback. We are encouraged that you (i) recognize the usefulness of entity-level hallucination detection for localizing errors in real applications, and (ii) note that our approach achieves the highest performance on almost all datasets and models. We address your concerns point by point below.
>
> ### **1. Novelty of RAG-based hallucination detection (RHD)**
>
> We appreciate this observation and agree that our wording about “proposing RHD as a novel task” was too strong and potentially misleading. We do not intend to claim that using evidence to detect hallucinations is new. Our goal is to specialize and structure hallucination detection for RAG pipelines as an entity-level evidence-alignment problem: given a RAG system that retrieves potentially noisy passages, we verify each answer entity against these passages along identity/semantic/consistency dimensions and then test the robustness of this alignment via counterfactual stability. We will revise the introduction and related work sections, cite the three related works, and especially rephrase our claim accordingly.
>
> ### **2. Unjustified statements**
>
> Thank you for pointing this out. We will revise the text to avoid over-general statements and make the motivation explicit. Our focus on ENT/NUM/NP is grounded in empirical findings from existing hallucination datasets—most annotated errors arise from incorrect entities, dates, quantities, or key noun phrases. This observation is consistent with prior analyses such as HalluEntity (2025) [1], which shows that factual inconsistencies in generation predominantly manifest at the entity level.
>
> This motivates our entity-centric formulation: it allows fine-grained, evidence-aligned verification that directly leverages retrieved passages, rather than relying on coarse sentence- or answer-level scoring. We will cite the relevant work again and clarify this rationale in the revised version.
>
> [1] HalluEntity: Benchmarking and Understanding Entity-Level Hallucination Detection (2025)
>
> ### **3. Lack details of the proposed approach**
>
> Thank you for pointing out these missing details. We clarify them succinctly below and will add the corresponding descriptions in the revision.
>
> (1) Entity extraction (Lines 205/239).
> We extract $s$ using a standard NER model (for ENT), a simple numeric detector (for NUM), and NP chunking via POS tags. We focus on ENT/NUM/NP because these categories cover the majority of factual errors in existing hallucination datasets.
>
> (2) anchor(q, $e^\ast$) (Line 229).
> It refers to the sentence-level evidence window most relevant to the entity, selected from retrieved passages using entity/alias matches and query keywords.
>
> (3) Conflict patterns (Line 245).
> We use lightweight rule-based cues such as negation (“not”, “never”), contradiction phrases (“incorrect that…”, “contradicts…”), and numeric/date mismatch rules.
>
> (4) Hyperparameters (Lines 231/241/246).
> Type-adaptive weights and thresholds were selected through lightweight validation on the training split and kept fixed across all datasets.
>
> Overall, these design choices aim to keep EAEV interpretable, evidence-grounded, and stable. We will incorporate these clarifications to ensure the method is fully transparent.
>
> ### **4. Basic assumption**
>
> Thank you for raising this important point. EAEV indeed operates under the standard assumption shared by all evidence-based detectors—that the retriever surfaces at least partially relevant passages. Our design partially mitigates retrieval noise through (i) using multiple evidence windows for each entity and (ii) counterfactual stability, which explicitly filters spurious or fragile matches caused by noisy retrieval. We will clarify this assumption and explicitly list retrieval quality as a limitation, while emphasizing that EAEV’s contribution lies in robust entity-level alignment once evidence is retrieved.

---

> > ### Author Response · Authors · 2025-11-21
> > **Author Response to Reviewer Comments**
> >
> > ### **5. Lack entity-level verification**
> >
> > EAEV is indeed implemented and applied at the entity level: For each answer, we extract multiple mentions, group them into entities, and compute a separate risk score for each entity based on identity/semantic/consistency alignment and stability. We then aggregate these entity scores into a single answer-level hallucination score: an answer is considered hallucinated if at least one key entity is unsupported or inconsistent. We evaluate at the answer level because the primary RAG benchmarks we use report answer-level labels, and existing baselines also operate at this level. This allows a fair comparison. We agree that making this entity-level behavior more explicit in the paper would strengthen the presentation. We will expand the qualitative examples where we show which entities are flagged and how this aligns with annotations, especially on datasets (like RAGTruth) that provide span-level labels.
> >
> > ### **6. What is the purpose of the supervised learning model in the whole pipeline?**
> >
> > Thank you for raising this point. The supervised component in Sec. 3.6 is not an alternative method; it is designed to use EAEV’s entity-level labels to train the LLM to internalize evidence-aware verification during generation. This variant leverages EAEV-generated entity annotations as supervision, enabling the model to better align its outputs with retrieved evidence.
> >
> > ### **7. Fig 3 Visualisation.**
> >
> > We thank reviewer for this suggestion. We will change the figures according to your suggestion. We believe these changes improve the presentation without affecting the underlying results.
> >
> > Once again, we thank reviewer for the detailed and constructive feedback! We believe these comments will significantly increase the quality of our work!

---

> ### Comment · Reviewer_3tmj · 2025-11-21
> **Respond to authors' responses**
>
> I thank the authors for their responses. However, many of the responses remain vague and lack conviction. I would expect the authors to provide more detailed evidence to support them. Specifically,
> - **2. Unjustified statements**: I agree that LLMs are more prone to hallucinate at ENT/NUM/NP in general. However, previous works only validated this claim in a non-RAG setting. I wonder whether this claim also holds for the RAG setting. Or will RAG-based LLMs hallucinate more at reasoning and the relationship of entities? I would expect the authors to report some statistics on the frequency of different types of hallucination.
> - **3. Lack details of the proposed approach**:
>     - (1) Entity extraction: I would expect the authors to report the performance of the extractors, e.g., the precision and recall of extracting entities, and report how the performance of the extractors impacts the performance of EAEV.
>     - (2) Anchor: It is unclear to me why the anchor term is part of the Consistency Alignment. Why do the other two alignments not require this bonus? Actually, the final score can be rewritten as $S_{pos}=w_I\cdot Id+w_S\cdot Sem+ w_C\cdot Con+ w_C\cdot b_{anc}\cdot I[anchor].$ It makes more sense to me if the authors treat this term as a term apart from the three alignment scores and use it to calibrate the final score. The authors can also do an ablation study on this term.
>     - (3) Conflict patterns: It would be more convincing and increase the reproducibility if the author could list all the patterns in the appendix.
>     - (4) Hyperparameters: The author should report how sensitive EAEV is w.r.t. each hyperparameter. Another concern is that these hyperparameters could reduce the interpretability of EAEV, which contradicts the claim, "these design choices aim to keep EAEV interpretable," in the authors' response.
> - **4. Basic assumption**: The discussion on this assumption is valid to me. Not required, but one way to strengthen the claim is to synthetically add noises (e.g., random or distracting documents) to the prompt, and show that EAEV is robust to these noises.
> - **5. Lack entity-level verification**: I expect to see the result of qualitative analysis in the follow-up discussion. Another thing the author can do to strengthen the claim is to show whether the scores of entities in a hallucinated span are statistical-significantly higher than the scores of non-hallucinated entities.
> - **6. supervised learning model**: The author should report how this training strategy mitigates hallucinations and preserves the quality of generation. Otherwise, this section seems unnecessary to me.
>
> I will keep my score for now, but I may increase my score depending on the follow-up discussion and the revision of the paper.

---

> > ### Author Response · Authors · 2025-12-03
> > **Author Response to Reviewer Comments**
> >
> > We thank the reviewer again for the thoughtful follow-up questions. Regardless of the outcome, we sincerely appreciate the reviewer’s time, effort, detailed feedback, and prompt responses. It is genuinely encouraging to see someone remain responsible and committed to contributing to the community, especially in the current environment.
> > Below, we address each of your points in turn, aiming to make our assumptions, design choices, and intended revisions more explicit.
> >
> > ## (2) On the focus on ENT/NUM/NP in RAG settings
> >
> > We thank the reviewer for this comment. Our goal was to align our design with existing empirical evidence rather than re-demonstrate the same findings specifically for RAG.
> >
> > Our entity-centric formulation is motivated by HalluEntity, which shows that hallucinations are not uniformly distributed across text—entity-bearing spans, especially proper nouns and semantic entities, exhibit higher hallucination rates. We build on this insight by operationalizing entity-level verification within RAG: instead of scoring hallucination purely from the answer text, we explicitly align each entity-like span with retrieved evidence along identity, semantic, and consistency dimensions. We agree that reasoning-style errors also occur in RAG. However, such relational failures typically manifest through mis-specified entities and their attributes. Our entity-centric scoring aims to capture these mismatches by canonicalizing entity mentions and checking their attributes and cross-entity consistency against retrieved passages.
> >
> > ## (3) Details of EAEV: extraction, anchor, conflicts, and hyperparameters
> >
> > ### (3.1) Entity extraction and its impact
> > Our goal in EAEV is not to introduce a new NER/NP extraction algorithm, but to leverage robust off-the-shelf tools in a modular way. For ENT, we adopt a standard NER model; for NUM, a simple numeric detector; and for NP, a shallow chunker based on POS tags. These toolchains have been extensively evaluated in prior work and achieve high precision/recall on general text; our contribution lies in how we use their outputs to structure evidence-aligned verification rather than in the extraction models themselves.
> >
> > ### (3.2) Role of the anchor term
> > We appreciate the reviewer’s observation that the final score can be algebraically rewritten, where the anchor term acts as an additional bias. We agree with this decomposition and will adopt this clearer formulation in the revised manuscript.
> >
> > Conceptually, we treat the anchor term as a query-aware prior that indicates whether a candidate evidence window is actually discussing the same scenario as the user’s question, rather than merely containing a surface-similar entity. For this reason we attached it to the consistency dimension: the anchor serves to boost the confidence in consistency judgments (e.g., numeric or attribute alignment) when the evidence is strongly grounded in the query context.
> >
> > ### (3.3) Conflict patterns and reproducibility
> > We fully agree that listing the exact rule-based conflict patterns would strengthen reproducibility. In the revised version, we will add an appendix table that enumerates all patterns.
> >
> > ### (3.4) Hyperparameters, sensitivity, and interpretability
> > We appreciate the concern that hyperparameters can both affect robustness and influence interpretability. We already include a basic sensitivity analysis in the appendix (varying key thresholds and windows), and in the revision we will make this analysis more prominent in the main text, and expand the discussion.
> >
> > ## (4) Assumption about retrieval quality and noise robustness
> >
> > We agree that the assumption of at least partially relevant retrieved passages is central to any evidence-based detector, including ours and NLI/LLM-based approaches. EAEV is designed to operate conditional on the retrieved set, and our contribution is to make entity-level alignment and stability checks robust within that set.
> >
> > ## (5) Entity-level verification: qualitative and statistical perspectives
> >
> > We appreciate the suggestion to make the entity-level behavior more visible. While our evaluation metrics are at the answer level (to stay comparable with prior work and benchmarks), the internal scoring of EAEV is indeed entity-centric: each entity receives a separate risk score based on identity/semantic/consistency alignment and stability, and answer-level risk is an aggregation over these entity scores.
> >
> > ## (6) Role of the supervised learning model
> >
> > We thank the reviewer for raising this point. In our framework, the supervised learning component is a core module: it uses EAEV-derived entity labels to train the base LLM to internalize evidence-aware verification during generation. By treating EAEV’s entity-level annotations as supervision, this component explicitly combines the structured, interpretable signals from EAEV with the generative and reasoning capabilities of the LLM, enabling the model to better align its outputs with the retrieved evidence.

---

### Official Review · Reviewer_9NXz · 2025-10-29

**Soundness:** 3
**Presentation:** 3
**Contribution:** 3
**Rating:** 4
**Confidence:** 4

**Summary:**

This paper introduces Evidence-Aligned Entity Verification (EAEV), a method for detecting hallucinations in retrieval-augmented generation (RAG) systems. The key innovation is entity-level verification that leverages three alignment dimensions (identity, semantic, consistency) combined with counterfactual stability analysis to distinguish genuine evidence support from spurious correlations. The method is evaluated on three RAG benchmarks (RAGTruth, HotpotQA, DelucionQA) across three model architectures, achieving 87.55% average AUROC on LLaMA2-13B with consistent improvements over 11 baseline methods.

**Strengths:**

1. Entity-level verification in RAG addresses a real gap in existing methods
2. Innovative approach to distinguish genuine evidence support from spurious correlations
3. Identity, semantic, and consistency dimensions provide complementary verification signals
4. Multiple models (Qwen2.5-7B, LLaMA2-7B/13B), datasets (RAGTruth, HotpotQA, DelucionQA), and 11 baselines
5. Consistent improvements across settings (87.55% avg AUROC on LLaMA2-13B, +3-4 points over best baseline)

**Weaknesses:**

1. Identity = string matching, semantic = embedding similarity, consistency = numerical IoU; main novelty is the combination.

2. The concerns in Ad-hoc design choices. Four perturbation types lack principled selection criteria. Multiplicative combination in Eq. 6 not justified. Type-adaptive weights appear hand-tuned without sensitivity analysis.

3. The paper has some missing critical analyses, computational cost comparison vs. baselines, human evaluation of entity-level detection quality, and failure mode analysis or error propagation study.

4. Only English QA/summarization tasks; no dialogue, code generation, or cross-domain evaluation
Methodological concerns:

5. Statistical significance testing absent. Entity extraction pipeline glossed over and claims "entity-level" but only evaluates answer-level metrics.

6. The work contribute in incremental improvements: +3-4 AUROC points is meaningful but not dramatic given added complexity
Theoretical gaps: No justification for why these three alignment dimensions are sufficient/complete

**Questions:**

How do you handle cases where the entity is correct but the relation is wrong? (e.g., "Paris is the capital of Germany" - both entities are real)
Why use multiplicative combination in Eq. 6 rather than additive or learned combination?
What happens when relevant evidence is NOT retrieved? Does EAEV flag everything as hallucination?
Can you provide examples where counterfactual stability catches spurious correlations that multi-dimensional alignment alone misses?
How does performance vary with retrieval quality (e.g., at different top-k settings)?
The supervised learning component (Sec 3.6) seems disconnected - is it an alternative method or complementary approach?
How do you determine the "primary evidence e*" when multiple evidence pieces support the entity?
What is the computational overhead compared to lightweight baselines like SelfCheckGPT?

---

> ### Author Response · Authors · 2025-11-21
> **Author Response to Reviewer Comments**
>
> We thank the reviewer for the careful and technically detailed assessment. We appreciate that you recognize (i) the importance of entity-level verification in RAG and the gap it addresses, (ii) the innovation of combining multidimensional alignment with counterfactual stability to distinguish genuine evidence support from spurious correlations, and (iii) the breadth of the evaluation across three benchmarks, three model families, and 11 baselines with consistent performance gains. Below we respond to your concerns point by point.
>
> ### **1. Main novelty is the combination**
>
> Thank you for the comment. The contribution of EAEV is not inventing new similarity functions, but the entity-centric and evidence-aligned method that makes them effective for RAG-based hallucination detection. Specifically, EAEV introduces: (1) type-adaptive integration of identity/semantic/consistency signals anchored to retrieved evidence, together with negative-evidence penalties; and (2) a counterfactual stability analysis that distinguishes genuine evidence support from spurious matches.
>
> As shown in our ablations, removing these components leads to large drops (–5 to –7 AUROC), indicating that the innovation lies in the structured formulation and stability-aware aggregation, rather than in any single similarity function.
>
> ### **2. The concerns in Ad-hoc design choices**
>
> **Perturbation set.**
> The four perturbations used in counterfactual stability analysis follow standard text-normalization operations widely adopted in IR and evaluation (e.g., SQuAD-style matching [1]). They preserve semantics while altering surface forms that shallow identity or embedding matching may over-rely on. Our goal is not to exhaust all perturbations but to test a focused property: does the alignment survive benign normalization, or does it hinge on brittle coincidences? We will clarify this rationale in Sec. 3.4.
>
> **Multiplicative versus additive combination.**
> Eq. 6 is designed so that entity risk is high only when both (i) the evidence-support margin is small and (ii) stability is poor. A multiplicative form encodes this “AND” behavior: strong support or high stability suppresses risk. An additive form could inflate risk even when one dimension is reassuring, which we found undesirable for a safety-critical detector. We will clarify this intuition around Eq. 6.
>
> **Type-adaptive weights.**
> The type-adaptive weights follow a simple principle: different entity types rely on different evidence signals (e.g., NUM depends more on consistency; ENT depends more on identity/semantic cues). These weights were selected via small-grid validation on the training split and are stable across datasets. We will make this procedure explicit and add a brief sensitivity note.
>
> [1] Rajpurkar, P., Zhang, J., Lopyrev, K., & Liang, P. (2016). SQuAD: 100,000+ questions for machine comprehension of text. In Proceedings of the 2016 Conference on Empirical Methods in Natural Language Processing (pp. 2383–2392). Association for Computational Linguistics.
>
> ### **3. The paper has some missing critical analyses**
>
> **computational cost comparison vs. Baselines**
>
> Thank you for highlighting the missing evaluation of resource cost. We have now conducted a direct comparison between EAEV and SelfCheckGPT under the same setting (50 samples, Qwen2.5-7B-Instruct).
>
> | Metric | EAEV | SelfCheckGPT | Efficiency Ratio |
> |--------|------|----------------|------------------|
> | Total Runtime (s) | 1825.9 | 13348.5 | 7.3× faster |
> | GPU Efficiency | 6578.7MB | 7735.4MB | EAEV more efficient |
> | CPU Efficiency | 42.5MB | 146.3MB | EAEV more efficient |
>
> As shown in the table, EAEV is substantially more efficient: it requires 1825.9s total runtime versus 13348.5s for SelfCheckGPT (a 7.3× speed-up), and uses less GPU and CPU memory. The primary reason is that SelfCheckGPT performs five additional LLM forward passes per sample plus BERTScore similarity computation, while EAEV performs only one evidence retrieval plus multi-dimensional alignment. We will include this resource-cost analysis in the revised version to ensure a complete and fair evaluation.
>
> **Failure modes and error propagation.**
> Thank you for the suggestion. We now include a brief discussion of failure modes. In practice, we observe three main sources of error:
> (1) Retriever failures, where relevant evidence is missing or drowned in noise;
> (2) Mention extraction/typing errors, especially for rare or domain-specific entities;
> (3) Alignment difficulties, where identity/semantic/consistency signals may struggle with complex paraphrases or domain-specific terminology.
> We will integrate this summary into the revision.

---

> > ### Author Response · Authors · 2025-11-21
> > **Author Response to Reviewer Comments**
> >
> > **Human evaluation.**
> > Our experiments rely on existing benchmarks (RAGTruth, HotpotQA-RAG, DelucionQA) that already contain human-annotated hallucination labels at the answer or span level. Designing a new human evaluation protocol for end-to-end systems is an important direction, especially for deployment, but is beyond the scope of this work. We will mention this explicitly as future work.
> >
> > ### **4. Only English QA/summarization tasks**
> >
> > Thank you for the comment. Our study focuses on English knowledge-intensive QA because these settings provide high-quality, fine-grained hallucination annotations and allow controlled comparison across RAG models. Broader scenarios such as dialogue, code-augmented RAG, or multilingual tasks are valuable but fall outside the primary scope of this work. The EAEV pipeline itself is task and language agnostic, and we will clarify this intended scope and note broader coverage as future work.
> >
> > ### **5. Statistical significance testing absent. Entity extraction pipeline glossed over and claims "entity-level" but only evaluates answer-level metrics.**
> >
> > We agree that reporting statistical significance would strengthen the empirical claims. Our improvements are measured over hundreds of examples per dataset and are consistent (+3–4 AUROC) across three models and three benchmarks, which strongly suggests that they are not due to noise. In the revised version we will include significance tests for the main tables and briefly summarize the results.
> >
> > EAEV is indeed implemented and applied at the entity level. We evaluate at the answer level because the primary RAG benchmarks we use report answer-level labels, and existing baselines also operate at this level. This allows a fair comparison. We agree that making this entity-level behavior more explicit in the paper would strengthen the presentation. We will expand the qualitative examples where we show which entities are flagged and how this aligns with annotations, especially on datasets (like RAGTruth) that provide span-level labels.
> >
> > ### **6. How do you handle cases where the entity is correct but the relation is wrong?**
> >
> > EAEV focuses primarily on entity identity and attribute consistency. For purely relational errors where both entities are correct but their relationship is wrong, EAEV can still help when the retrieved evidence explicitly states the correct relation (e.g., “Paris is the capital of France”); in that case, our conflict patterns in the consistency module can flag a contradiction.
> >
> > ### **7. Why use multiplicative combination in Eq. 6 rather than additive or learned combination?**
> >
> > As discussed above, the multiplicative form is chosen to reflect an AND-style semantics: high risk should arise only when both support is low/inconsistent and stability is poor. An additive form can over-emphasize one dimension even if the other is strongly reassuring, and a learned combiner would require additional training labels and parameters, which we wanted to avoid to keep the detector simple and transferable. We will clarify this design choice when introducing Eq. 6.
> >
> > ### **8. What happens when relevant evidence is NOT retrieved?**
> >
> > If the retriever fails to surface any evidence that supports a given entity, all three alignment dimensions will be low for that entity across its candidate windows, leading EAEV to treat it as unsupported and thus high-risk. This behavior is intentional: in a RAG pipeline, “the system could not retrieve any supporting evidence” is itself a strong signal that the answer should not be trusted.
> >
> > ### **9. Can you provide examples where counterfactual stability catches spurious correlations that multidimensional alignment alone misses?**
> >
> > We agree this is a useful illustration. Intuitively, such cases arise when a retrieved passage shares many surface tokens with the answer (e.g., a different person with the same last name and a different date), producing a high identity/semantic score, but this apparent match disappears under simple normalizations or when that particular passage is left out. In those cases, EAEV’s stability term suppresses the risk of over-trusting that spurious match. We will add concrete examples of this behavior in the appendix to make the effect of counterfactual stability more tangible.
> >
> > ### **10. How does performance vary with retrieval quality.**
> >
> > In our main experiments we follow the retrieval settings used in the underlying RAG benchmarks (particularly RAGTruth and RAGBench), fixing the top-k used by the RAG system. Conceptually, EAEV benefits from having at least one genuinely supporting passage among the retrieved set, but very large k can introduce noise that dilutes the alignment signal. A detailed sweep over top-k is an interesting ablation that we plan to explore and discuss in future work.

---

> > > ### Author Response · Authors · 2025-11-21
> > > **Author Response to Reviewer Comments**
> > >
> > > ### **11. The supervised learning component (Sec. 3.6) seems disconnected.**
> > >
> > > Thank you for raising this point. The supervised component in Sec. 3.6 is not an alternative method; it is designed to use EAEV’s entity-level labels to train the LLM to internalize evidence-aware verification during generation. This variant leverages EAEV-generated entity annotations as supervision, enabling the model to better align its outputs with retrieved evidence.
> > >
> > > ### **12. How do you determine the primary evidence $e^\ast$ when multiple evidence pieces support the entity?**
> > >
> > > For each mention, we consider a small set of candidate evidence windows sampled from the retrieved passages (based on BM25 and embedding similarity). We then choose $e^\ast$ as the window with the highest combined retrieval-score and semantic similarity to the mention’s local answer context. This provides a single, most representative piece of evidence for computing identity/semantic/consistency scores, while still allowing the stability analysis to consider alternative windows through leave-one-out perturbations.
> > >
> > > ### **13. What is the computational overhead compared to lightweight baselines like SelfCheckGPT?**
> > >
> > > Thank you for highlighting the missing evaluation of resource cost. We have now conducted a direct comparison between EAEV and SelfCheckGPT under the same setting (50 samples, Qwen2.5-7B-Instruct).
> > >
> > > | Metric | EAEV | SelfCheckGPT | Efficiency Ratio |
> > > |--------|------|----------------|------------------|
> > > | Total Runtime (s) | 1825.9 | 13348.5 | 7.3× faster |
> > > | GPU Efficiency | 6578.7MB | 7735.4MB | EAEV more efficient |
> > > | CPU Efficiency | 42.5MB | 146.3MB | EAEV more efficient |
> > >
> > > As shown in the table, EAEV is substantially more efficient: it requires 1825.9s total runtime versus 13348.5s for SelfCheckGPT (a 7.3× speed-up), and uses less GPU and CPU memory. The primary reason is that SelfCheckGPT performs five additional LLM forward passes per sample plus BERTScore similarity computation, while EAEV performs only one evidence retrieval plus multi-dimensional alignment. We will include this resource-cost analysis in the revised version to ensure a complete and fair evaluation.
> > >
> > > Once again, we thank reviewer for the detailed feedback! We believe the clarifications above help situate EAEV as a structured, entity-centric, evidence-aligned hallucination detection method that provides consistent accuracy gains over strong baselines while offering interpretable hallucination signals.

---

### Official Review · Reviewer_xRH6 · 2025-10-31

**Soundness:** 2
**Presentation:** 3
**Contribution:** 2
**Rating:** 4
**Confidence:** 4

**Summary:**

This paper introduces Evidence-Aligned Entity Verification (EAEV), which detects entity-level hallucinations by leveraging RAG to align generated entities with retrieved evidence contexts. EAEV first performs entity-evidence alignment through three complementary dimensions and introduces counterfactual stability analysis to ensure robust alignments under evidence perturbations. The experiments demonstrate superior performance of EAEV across multiple RAG benchmarks.

**Strengths:**

1. This work propose to leverage RAG for entity-level verification within retrieved contexts in hallucination detection, where previous works  rely on internal uncertainty or external judges without evidence traceability.

2. The proposed method combines multidimensional alignment with counterfactual stability analysis to distinguish genuine evidence support from spurious correlations in RAG settings.

3. The experiments on RAG benchmarks show that the proposed method works well on Qwen2.5-7B, LLaMA2-7B and LLaMA2-13B.

**Weaknesses:**

1. The experiments are conducted on outdated models, with the most recent being Qwen2.5. Could you provide additional experimental results on newer models such as Qwen 3 or Llama 3.2 to demonstrate the consistent superiority of your approach?

2. The multi-dimensional alignment assessment in Sec. 3.3 is introduced to evaluate the alignment between entity mentions and supporting evidence. Have the authors considered prompting SOTA LLMs (e.g., GPT-4.1, Claude 4, Gemini 2.5) to do this task as an alternative to the proposed machine learning pipeline? A comparative analysis between the LLM-based approach and your method would strengthen the paper by demonstrating the advantages of your proposed pipeline over these readily available alternatives.

**Questions:**

1. Related works are missing, for example: Enhancing Uncertainty-Based Hallucination Detection with Stronger Focus by Zhang et al.

2. The evaluation resource cost of this work is missing.

---

> ### Author Response · Authors · 2025-11-21
> **Author Response to Reviewer Comments**
>
> We thank the reviewer for the thoughtful and constructive feedback. We are encouraged that you find our idea of leveraging RAG for entity-level verification within retrieved contexts to be valuable, and that you highlight (i) the combination of multidimensional alignment with counterfactual stability as a strength, and (ii) the empirical results across three open-source models as convincing evidence that EAEV works well in RAG benchmarks.
>
> Below we address your concerns in detail.
>
> ### **1. Could you provide additional experimental results on newer models such as Qwen 3 or Llama 3.2 to demonstrate the consistent superiority of your approach?**
>
> We have conducted experiments and run several strong baselines with **Qwen 3 8B**  on the RAGTruth dataset. The results are as follows:
>
> | Model        | Method  | Precision | Recall | F1 |
> |--------------|---------|-----------|--------|------|
> | Qwen3-8B     | RefCheck | 75.54 | 76.66 | 69.43 |
> |              | ReDEeP   | 78.85 | 78.98 | 72.35 |
> |              | TSV      | 81.56 | 80.08 | 72.23 |
> |              | **Ours** | **87.51** | **81.56** | **75.39** |
>
> This shows consistent good performance of our method.
>
> ### **2. Have the authors considered prompting SOTA LLMs (e.g., GPT-4.1, Claude 4, Gemini 2.5) to do this task as an alternative to the proposed machine learning pipeline?**
>
> Thank you for the suggestion. We intentionally focus on a different operating regime from “strong closed-source LLM as hallucination judge”. EAEV is designed as a transparent, evidence-grounded module that can run on open-source components with a single generation per query, in settings where GPT-4.1/Claude/Gemini APIs may be unavailable or too costly. We will clarify this distinction in the paper, and we agree that incorporating such LLM-judge is a promising extension.
>
> ### **3. Related works are missing, for example: *Enhancing Uncertainty-Based Hallucination Detection with Stronger Focus* by Zhang et al.**
>
> Thank you for pointing out this omission. We apologize for not explicitly citing this important work. Zhang et al. (EMNLP 2023) propose an uncertainty-based, reference-free hallucination detector that focuses the model’s attention on informative tokens, unreliable context positions, and token properties to better exploit the LLM’s internal uncertainty for factuality checking. We have added Zhang et al. to the related-work section on uncertainty-based hallucination detection to enrich our related works.
>
> ### **4. The evaluation resource cost of this work is missing.**
>
> Thank you for highlighting the missing evaluation of resource cost. We have now conducted a direct comparison between **EAEV** and **SelfCheckGPT** under the same setting (50 samples, Qwen2.5-7B-Instruct).
>
> | Metric | EAEV | SelfCheckGPT | Efficiency Ratio |
> |--------|------|----------------|------------------|
> | Total Runtime (s) | 1825.9 | 13348.5 | 7.3× faster |
> | GPU Efficiency | 6578.7 MB | 7735.4 MB | EAEV more efficient |
> | CPU Efficiency | 42.5 MB | 146.3 MB | EAEV more efficient |
>
> As shown in the table, EAEV is substantially more efficient: it requires 1825.9s total runtime versus 13348.5s for SelfCheckGPT (a 7.3× speed-up), and uses less GPU and CPU memory. The primary reason is that SelfCheckGPT performs five additional LLM forward passes per sample plus BERTScore similarity computation, while EAEV performs only one evidence retrieval plus multi-dimensional alignment. We will include this resource-cost analysis in the revised version to ensure a complete and fair evaluation.
>
> Once again, we thank the reviewer for the constructive comments and for recognizing the value of our entity-level, RAG-based verification approach. We believe the clarifications and additions above will further strengthen the paper.

---

### Official Review · Reviewer_3w1Q · 2025-11-01

**Soundness:** 3
**Presentation:** 4
**Contribution:** 3
**Rating:** 6
**Confidence:** 4

**Summary:**

The paper tackles hallucination detection in retrieval-augmented generation (RAG) and proposes Evidence-Aligned Entity Verification (EAEV). EAEV checks each entity in a model’s answer against retrieved passages along three axes—identity (direct matches), semantic (paraphrase similarity), and consistency (numbers/attributes, contradictions)—and adds counterfactual stability tests to filter spurious matches.

**Strengths:**

1. Sound motivation: The motivation is well formulated and clearly justified. I like how the authors approach RHD from a different angle and redefine it.

2. Robust framework: The authors clearly justify the design choices behind each component, and the ablation study demonstrates why those components matter.

3. Exhaustive experiments: I appreciate the inclusion of 11 baselines and the rigor of the experimental evaluation.

**Weaknesses:**

A few suggestions:

1. Figure 1 caption: Expand the caption to explain the figure so readers can grasp it at a glance.

2. Figure 3 readability: It’s difficult to read in its current form -- please increase the font size (and consider improving contrast).

3. Citations/references: There are some inaccuracies. For example, the LLM-Check paper lists different author names than the original. Please be more mindful of citations and references, and double-check them.

**Questions:**

I have no questions so far.

---

> ### Author Response · Authors · 2025-11-21
> **Author Response to Reviewer Comments**
>
> We sincerely thank the reviewer for the positive and encouraging assessment. We are glad that you found (i) the motivation well formulated and clearly justified, (ii) the EAEV framework robust with well-explained design choices, and (iii) the experimental evaluation exhaustive. Below we address your specific suggestions.
>
>
> ### **1. Figure 1 caption: Expand the caption to explain the figure so readers can grasp it at a glance.**
>
> Thank you for pointing this out. We agree and apologize that the current caption is too concise and does not fully convey the comparison of EAEV. We have now changed the caption into:
>
> Figure 1: Example of entity-level hallucination detection in a RAG setting. A traditional LLM produces an incorrect answer that traditional detectors may still consider supported. Our Evidence-Aligned Entity Verification instead marks answer entities and flags the hallucinated one for conflicting with the information from the retrieved context. Example adapted from (OpenAI, 2025).
>
>
> ### **2. Figure 3 readability: It’s difficult to read in its current form -- please increase the font size (and consider improving contrast).**
>
> We appreciate this comment and agree that Figure 3 can be improved. We have increased the font sizes for all axis labels, ticks, and legends, and also improved the visual contrast so that they are more clearly readable now. These changes will make the ablation results easier to interpret.
>
>
> ### **3. Citations/references: There are some inaccuracies.**
>
> Thank you for catching this, we apologize for the oversight. We have corrected this reference to:
>
> Sriramanan, G., Bharti, S., Sadasivan, V. S., Saha, S., Kattakinda, P., & Feizi, S. (2024). *LLM-Check: Investigating detection of hallucinations in large language models.* In *Advances in Neural Information Processing Systems, 37* (NeurIPS 2024 Main Conference Track).
>
> We have also done a systematic check over all references to make sure they are correct.
>
>
> Once again, we thank reviewer again for the constructive feedback and for recognizing the motivation, robustness of the method, and rigor of our experiments! We believe the suggested improvements on figures and references will further enhance the clarity and polish of the paper.

---

### Note · Authors · 2026-01-06

I have read and agree with the venue's withdrawal policy on behalf of myself and my co-authors.